# Prostate Ultrasound Image Segmentation Based on DSU-Net

**DOI:** 10.3390/biomedicines11030646

**Published:** 2023-02-21

**Authors:** Xinyu Wang, Zhengqi Chang, Qingfang Zhang, Cheng Li, Fei Miao, Gang Gao

**Affiliations:** 1College of Information Science and Technology, Northwest University, Xi’an 710127, China; 2College of Computer Science and Technology, Xidian University, Xi’an 710071, China; 3Xi’an Key Laboratory of Human-Machine Integration and Control Technology for Intelligent Rehabilitation, Xijing University, Xi’an 710123, China; 4Department of Ultrasound, Ruijin Hospital, Shanghai Jiao Tong University School of Medicine, Shanghai 200025, China; 5Shanghai Yiran Health Consulting Co., Ltd., Shanghai 201821, China

**Keywords:** prostate ultrasound, deformable convolution, shear U-Net, image segmentation

## Abstract

In recent years, the incidence of prostate cancer in the male population has been increasing year by year. Transrectal ultrasound (TRUS) is an important means of prostate cancer diagnosis. The accurate segmentation of the prostate in TRUS images can assist doctors in needle biopsy and surgery and is also the basis for the accurate identification of prostate cancer. Due to the asymmetric shape and blurred boundary line of the prostate in TRUS images, it is difficult to obtain accurate segmentation results with existing segmentation methods. Therefore, a prostate segmentation method called DSU-Net is proposed in this paper. This proposed method replaces the basic convolution in the U-Net model with the improved convolution combining shear transformation and deformable convolution, making the network more sensitive to border features and more suitable for prostate segmentation tasks. Experiments show that DSU-Net has higher accuracy than other existing traditional segmentation methods.

## 1. Introduction

With the acceleration of the modernization process, people’s quality of life has been greatly improved in all aspects, but the increasing pressure of life and work has also caused a certain threat to health. The prostate is a substantial gland located at the root of the urethra that surrounds the urethra and has a slightly flattened chestnut shape. Prostate cancer is one of the most common cancers among men [1,2], and its incidence increases with age, which seriously affects the health of men [3]. Since there are no obvious symptoms in the early stage of prostate cancer, many prostate cancer patients are already in the advanced stage of cancer at the first visit, which easily delays the best treatment period. In traditional diagnosis, doctors diagnose the disease by asking about symptoms, looking at and touching the affected area, etc., but most patients cannot accurately describe the symptoms. It is worth noting that prostate cancer patients have no obvious symptoms in the early stage. Therefore, this will cause doctors to make deviations in diagnosis, which also brings great challenges to the treatment of prostate cancer in China [4]. With the continuous updating of medical equipment, the emergence of transrectal ultrasound (TRUS), magnetic resonance imaging (MRI), computed tomography (CT) and other medical images can allow doctors to intuitively see the distribution and morphology of internal organs and lesion areas in the human body.

TRUS-guided biopsies are one of the most commonly used and effective methods for diagnosing of prostate cancer [5,6], but a biopsy will bring physical and psychological pain to patients [7]; it cannot cover the entire prostate gland which is prone to missed diagnosis [8]. In some countries and regions, the efficiency of examination is low, and patients need to make an appointment and wait for a prostate needle biopsy, which will delay the timely treatment of patients [9]. Due to the noninvasive and low-cost characteristics of TRUS, the development of prostate gland lesions can be observed in real time. During the diagnosis and treatment of prostate cancer, doctors need to view the TRUS image of prostate in real time (as shown in Figure 1). For example, the position and boundary of the prostate need to be accurately located in the biopsy puncture and surgical treatment; that is, the prostate region in the image should be segmented. In clinical practice, doctors usually need to manually segment the prostate, but due to the unclear edge of the prostate TRUS image, manual segmentation will take a lot of time and labor costs, and the accuracy of the needle biopsy and prostate segmentation will also be affected by doctors’ subjective experience.

Computer vision technology (especially deep neural network models) in the field of image processing and deep learning, which is in continuous development, has been widely used in organ segmentation tasks to assist doctors in diagnosis. However, at present, most segmentation deep learning algorithms are aimed at medical images such as CT and MRI with good image quality, while there is relatively little research on segmentation methods for ultrasound images with poor quality based on deep learning. In recent years, scholars have proposed some ultrasound image segmentation methods based on deep learning. Among them, Yang et al. [10] proposed a method which uses a recurrent neural network RNN in sequential ultrasound images to learn shape priors and combine multiscale features for organ segmentation. The experimental results showed that the prostatic segmentation with an incomplete boundary was better than the traditional network; Mishra et al. [11] put forward a full-convolutional neural network with attention depth supervision for the automatic and accurate segmentation of blood vessels in liver ultrasound images. In the split test of the public dataset, the accuracy rate was higher than that of networks such as FCN. In the image segmentation task, these algorithms obtained good segmentation results, but the segmentation effect on the asymmetrical, irregular, blurred, and difficult-to-distinguish prostate in TRUS images was relatively poor. This is due to the interaction between the ultrasound and tissue, and the ultrasound image is affected by various factors, such as damage to human tissue and probes, resulting in the insufficient clarity of the prostate TRUS image. It can be observed that the border of the prostate in the TRUS image is not clear, and the shape difference is large, irregular, asymmetric, etc., which have brought challenges to prostate segmentation in TRUS images.

Accurately segmenting the prostate in TRUS can also guide physicians in the diagnosis and surgical treatment of prostate cancer. It is impractical that large-scale manual labeling for segmentation requires a lot of time and labor costs. Therefore, the prostate segmentation algorithm based on TRUS images can automatically segment the prostate region of TRUS images, which obtains more accurate segmentation results.

### Related Work

The fully convolutional network (FCN) was proposed by Jonathan et al. [12] in 2015. Images of any size can be sent as inputs to the FCN model for training. The up-sampling operation restores the feature map obtained by the last convolution to the same size as the input image. These operations not only save the spatial information of the input image but also realize the prediction of each pixel, so as to obtain the image segmentation result. However, when the FCN model restores the features to the original image size, there will be deviations in pixel positioning, and pixel predictions with inaccurate positions will also have a certain impact on the segmentation results [13].

Ronneberger et al. [14] proposed U-Net, an improved image full-convolution semantic segmentation network model based on FCN, which achieved excellent segmentation results in medical image segmentation tasks and was widely applied in various segmentation tasks. Similar to the FCN network structure, the U-Net network also includes a down-sampling stage and an up-sampling stage. However, the difference between the two is that the down-sampling stage of the U-Net network is symmetrical to the up-sampling stage; that is, in the two stages, the number of convolution operations is the same. Moreover, adding a skip connection structure combines the feature map obtained by the down-sampling layer with that corresponding to the feature map connection obtained by the up-sampling layer, which makes the skip connection realize the fusion structure of the feature. In the encoder part of the network, the convolution operation and down-sampling operation reduce the size of the feature graph when extracting features. In the decoder part, the features extracted by the lower sampling layer are restored to the size of the initial feature map through a deconvolution operation; the lower sampling layer and the upper sampling layer are connected by a skip connection to improve the accuracy of pixel positioning in the network, thereby achieving a more accurate segmentation effect. Therefore, U-Net is also more suitable for medical image segmentation tasks.

At present, the U-Net network model is widely used in the semantic segmentation tasks of various medical images, and its segmentation performance is excellent, but in the segmentation tasks of prostate TRUS images, there will still be certain traditional network segmentation problems. The common problem is that the accuracy of the segmentation results for prostates with asymmetrical, irregular, and blurred edges is not high enough. Therefore, in response to these problems, the DSU-Net prostate segmentation method proposed in this paper is based on an improvement in the U-Net network model structure, and the segmentation effect obtained by the improved model has been improved.

## 2. Prostate Segmentation Method Based on DSU-Net

### 2.1. Data Preprocessing

Convolutional neural network training requires a certain amount of data as the basis; sufficient prostate TRUS image data and corresponding labeling data are the necessary prerequisites for prostate segmentation using computer vision technology. Before data preprocessing, we made sure of the following three things: (1) all samples were of the same size as well as the number of channels; (2) the images were eventually to be fed to the convolutional network as a tensor; (3) normalization was performed for each channel of each image. These three steps were designed to make the convolutional neural network run smoothly and to make the training process smoother.

In the preprocessing stage, due to the small number of publicly available prostate TRUS image datasets, one is limited by issues such as patient privacy. Therefore, the data augmentation of prostate TRUS datasets is performed with the aim of helping the network to train properly and improve the robustness to avoid overfitting [15,16]. Furthermore, the data enhancement strategies include random direction translation, clipping, and horizontal flip [17,18]. These operations not only correct the random errors of the image acquisition system and the instrument position (such as the imaging angle, perspective relationship, or even the lens itself), but also enrich the amount of image information, enhance the image interpretation and recognition, and meet the needs of some special analysis. The prostate region in the TRUS image is shown in Figure 2.

### 2.2. Prostate Segmentation Method Based on DSU-Net

In the convolution operation of the same layer of the CNN, the receptive fields of all activation units are the same, and the convolution kernels of the same size always sample the input features at a fixed position and perform pooling operations on the feature map at a fixed ratio. The formula expression of the traditional convolution structure is shown in Equation (1), where R defines the size and expansion of the receptive field with a 3 × 3 convolution kernel of expansion size 1. Formula (2) represents the calculation of each position $M_{0}$ in the output feature map y after convolution (the calculation can be described as the summation of the weighted sample values to obtain the output feature map after sampling the input map χ using a 3 × 3 convolution kernel of expansion size 1), where $M_{n}$ denotes the set of all positions in R, χ denotes the input feature map, and w denotes the weight of the sampled values at that point.
(1)$$R=\left\{\begin{array}{c} \left(-1,-1\right),\left(-1,\;0\right),\left(-1,\;1\right),\\ \left(0,-1\right),\left(0,\;0\right),\left(0,\;1\right),\\ \left(1,-1\right),\left(1,\;0\right),\left(1,1\right) \end{array}\right\}$$
(2)$$y\left(M_{0}\right)=\underset{M_{n}\in R}{\sum }w\left(M_{n}\right)\cdot x\left(M_{0}+M_{n}\right)$$
where

Deformable convolution [19] is an operation that adds an offset to the traditional fixed convolution without being confined to the underlying convolution frame, thus focusing more on the region of interest and enhancing the network’s ability to adapt to object deformation. Therefore, visual features of different shapes and scales can be captured, adapted, and extracted by deformable convolution as well as receptive fields. The segmentation network in this paper was obtained by the following steps: firstly, replacing the normal fixed convolution in the U-Net model with the deformable convolution; secondly, adding the shear transform and its inverse transform before and after the deformable convolution operation which was to be obtained.

The calculation method of each position $M_{0}$ in the output feature maps y obtained after the deformable convolution operation is shown in Formula (3), where $\Delta M_{n}$ is the offset of the irregular sampling position relative to $M_{0},$ which is usually a decimal. The bilinear interpolation method is subsequently applied to calculate the eigenvalues of the output.
(3)$$y\left(M_{0}\right)=\underset{M_{n}\in R}{\sum }w\left(M_{n}\right)\cdot x\left(M_{0}+M_{n}+\Delta M_{n}\right)$$

Figure 3 shows a schematic of the base convolution and deformable convolution sampling with a convolution kernel of size 3 × 3.

Shear transformation [20] (ST) is a widely used linear transformation method, which has been applied in many tasks such as image denoising, sparse image representation, and edge detection. Most images contain a lot of linear information, and extracting edges, line segments, and other features involved in image processing and recognition plays an important role in the results of the task. However, because the image contains a complex background, the extracted image information will be incomplete, and the introduction of shear transform can detect more image edge information and obtain more edge features.

Shear transformation includes the horizontal direction and vertical direction. The horizontal direction is denoted as $S_{shear\_h}$, the vertical direction is denoted as $S_{shear\_v}$, and the shear transformation can be expressed as $S_{shear}$ = [$S_{shear\_h}$,$S_{shear\_v}$]. where $S_{shear\_h}=\left[\begin{array}{ll} 1 & 0\\ \frac{\left\lfloor k\right\rfloor }{2^{ndir}} & 1 \end{array}\right]$, $S_{shear\_v}=\left[\begin{array}{ll} 1 & \frac{\left\lfloor k\right\rfloor }{2^{ndir}}\\ 0 & 1 \end{array}\right]$, ndir is the direction parameter, k∈N and $k\in [{\left(-2\right)}^{ndir},2^{ndir}]$.

The image transformation formula for when the direction is horizontal is shown in Equation (4):(4)$$\left(x_{h}^{\prime },y_{h}^{\prime })=(x,y)\times S_{shear\_h}=(x,y)\times \left[\begin{array}{ll} 1 & 0\\ \frac{\left\lfloor k\right\rfloor }{2^{ndir}} & 1 \end{array}\right]=(x+y\times \frac{\left\lfloor k\right\rfloor }{2^{ndir}},y\right)$$

The image transformation formula for when the direction is vertical is shown in Formula (5):(5)$$\left(x_{h}^{\prime },y_{h}^{\prime })=(x,y)\times S_{shear\_v}=(x,y)\times \left[\begin{array}{ll} 1 & \frac{\left\lfloor k\right\rfloor }{2^{ndir}}\\ 0 & 1 \end{array}\right]=(x,x\times \frac{\left\lfloor k\right\rfloor }{2^{ndir}}+y\right)$$
where (x,y) is the coordinate point of the element in the original picture, and $\left(x_{h}^{\prime },y_{h}^{\prime }\right)$ is the coordinate point of the element in the image after the shear transformation is applied. The element value of the coordinate point remains unchanged before and after the transformation.

The image can be restored to the original image by applying its inverse transformation. The inverse shear transformation also includes horizontal and vertical directions, and the horizontal direction is denoted as $S_{ishear\_h}$, the vertical direction can be expressed as $S_{ishear\_v}$, and the inverse shear transformation can be expressed as $S_{ishear}$ = [$S_{ishear\_h}$,$S_{ishear\_v}$]. where $S_{ishear\_h}=\left[\begin{array}{cc} 1 & 0\\ -\frac{\left\lfloor k\right\rfloor }{2^{ndir}} & 1 \end{array}\right]$, $S_{ishear\_v}=\left[\begin{array}{cc} 1 & -\frac{\left\lfloor k\right\rfloor }{2^{ndir}}\\ 0 & 1 \end{array}\right]$.

In this paper, the shear transform with lateral parameters of 0.3 and 0.6 (shown in Figure 4b,c); Figure 4a is the original image) is introduced to better fit the edge information of the boundary during image feature extraction, so as to complete the segmentation of the prostate in the TRUS images.

The DSU-Net model proposed in this paper replaces the original 3 × 3 convolution with the deformable convolution and performs shear transformation and inverse shear transformation before and after the deformable convolution. Firstly, the shear transformation is added before the regular 3 × 3 convolution; secondly, bias learning of convolution is added to generate a deformable convolution kernel with irregular shape; and finally, the inverse shear transformation is added and, then, the feature map is restored to the original size so that the next convolution layer can extract the image features. At the stage of the encoder and decoder, the model fits the boundary information of different scales based on deformable convolution by learning local adaptive receptive fields. During the training process, the deformable convolution obtains the offset through a convolution operation on the same input feature map, and then simultaneously learns and obtains the convolution kernel and offset used to generate the output feature map.

The prostate segmentation network based on DSU-Net is shown in Figure 5. After the initial feature map is sent to the network, high-dimensional features are obtained through the down-sampling operation of the feature extraction network on the left; then, four up-sampling operations of the feature fusion network are used to restore the original image size, and the high-dimensional features and low-dimensional features of the same layer are spliced by skip connections. Finally, the segmentation image is obtained through the convolution layer of 1 × 1 and the softmax layer. Each 3 × 3 convolution in the model incorporates the improved convolution of shear transformation, deformable convolution, and inverse shear transformation, using the binary cross entropy function as a loss function to train the segmentation network.

## 3. Experiment and Analysis

The experiments in this paper were created, trained, and tested on the Pytorch platform. The hardware of the experimental environment are as follows. CPU: Intel(R) Core(TM) I7-7800X CPU @ 3.50 ghz, GPU: GeForce GTX 2080Ti, operating system: 64-bit, and memory: 64 GB. The software information is Ubuntu 16.04, PYTHon3.6.2, pyTorch1.1.0. The specific experimental parameters are shown in Table 1.

### 3.1. Dataset

In order to verify that the method proposed in this paper can effectively segment the prostate, all of our data were provided by Shanghai Ruijin Hospital and were obtained by doctors using specialized medical equipment to obtain ultrasound images of the prostate. We acquired a total of 1057 prostate TRUS images with a resolution of 800 × 608. All data were guided by professional radiologists, using 3DSlicer medical image processing tools to mark the prostate area of TRUS images, and then checked and confirmed by professional radiologists. In this paper, the training data are 780 cases, the verification data are 123 cases, and the test data are 154 cases. In the experiment, the TRUS image and the corresponding label were cropped to an image size of 581 × 477 which was the prostate image after removing part of the prostate image information.

### 3.2. Evaluation Index

In this paper, two quantitative indexes are used to analyze the results of prostate segmentation, which are the commonly used Dice coefficient and Jaccard similarity coefficient.

The Dice coefficient is defined as the similarity or overlap degree of two samples, which is a calculation function to evaluate the similarity between samples. The calculation formula is as follows:(6)$$Dice=\frac{2\left|\;GT\cap SR\;\right|}{\left|\;GT\;\right|+\left|\;SR\;\right|}$$

GT is the real label of the image and SR is the actual segmentation result obtained by the model. GT∩SR represents the overlap between the real labels and the actual segmentation results of the model.

For the calculation of Dice coefficients (as shown in Equation (6)):

First, |GT∩SR| is approximated as a dot product between the prediction graph (SR) and label (GT), and the results of the elements of the dot product are summed; secondly, for the binary classification problem, the GT partition graph is only two values of 0, 1, so |GT∩SR| can effectively zero out all pixels that are not activated in the GT partition graph in the prediction partition graph. For the activated pixels, the main penalty is the low-confidence prediction, and higher values will give better Dice coefficients. The above can be summarized as: (1) The pointwise multiplication of the predicted partition map with the GT partition map; (2) the summation of the resultant elements of the element-by-element multiplication; (3) computation of |GT| and |SR|, where either direct element-wise summation or element-wise summation can be used.

The Jaccard coefficient is used to compare similarities and differences between finite sample sets, that is, the intersection ratio of two sets. The numerator is the intersection of two finite sample sets, and the denominator is the union of two finite sample sets.

The Jaccard coefficient is calculated as follows: the metric related to the Jaccard coefficient is called the Jaccard distance, which is used to describe the dissimilarity between sets. The larger the Jaccard distance, the less similar the sample is.

The Jaccard coefficient calculation formula is as follows (the meaning of GT and SR has been explained in this chapter):(7)$$Jaccard=\frac{\left|\;GT\cap SR\;\right|}{\left|\;GT\cup SR\;\right|}=\frac{\left|\;GT\cap SR\;\right|}{\left|\;GT\;\right|+\left|\;SR\;\right|-\left|\;GT\cap SR\;\right|}$$

### 3.3. Experimental Results

In this section, the DSU-Net segmentation method that combines shear transformation and deformable convolution to segment the prostate in TRUS images is proposed. The results were compared with the original DeepLabv3+ [21] network model, the original PSPNet [22] network model, the PSPNet network model with the shear transform, the original U-Net network model, and the U-Net network model with the shear transform; the variable segmentation results obtained by the DU-Net network model of the convolution unit are compared to verify the effectiveness of the proposed method in terms of quantitative indicators and visual results. In short, the values obtained from the evaluation metrics (Dice and Jaccard in Section 3.2) are compared to verify the validity of the methods in this paper.

The results of prostate segmentation obtained with this method are shown in Table 2. As can be seen from the results listed in the table, compared with the traditional basic U-Net network, the effect of adding deformable convolution and shear transformation has been improved to a certain extent. The experimental results obtained with the method in this paper are closest to the real label, which proves the effectiveness of the improved algorithm in this paper.

Figure 6 shows the visual comparison of the TRUS segmentation results of the prostate in this paper. Figure 6a shows the original prostate TRUS image input into the segmentation network; Figure 6b shows the actual labels corresponding to the TRUS images of the prostate, with black as the background and white as the prostate region to be segmented; Figure 6c shows the segmentation results of the tested prostate TRUS images on the Deeplabv3+ model; and Figure 6d shows the segmentation results of the tested prostate TRUS images on the Deeplabv3+ model with shear transformation. Figure 6e shows the segmentation results of the tested prostate TRUS images on the PSPNet model; Figure 6f shows the segmentation results of the tested prostate TRUS images on the PSPNet model with shear transformation; and Figure 6g shows the segmentation results of the tested prostate TRUS images on the U-Net model. Figure 6h is the segmentation result of the tested prostate TRUS image on the U-Net model with shear transformation; Figure 6i is the segmentation result of the tested prostate TRUS image on the DU-Net model with a deformable convolution module; and Figure 6j is the prostate segmentation result obtained with the proposed method in this paper.

It can be seen from Figure 6 that the segmentation results of DSU-Net are very close to the ground truth. In the third image, the prostate gland is missing in the lower-right corner compared with the real area in the segmentation result of the comparison network. Compared with the experimental comparison network, the missing part in the method in this paper is less, which is closer to the truth value label. The segmentation results obtained with the proposed method can segment the prostate more accurately and present more precise segmentation results than other methods.

## 4. Conclusions

In this work, we proposed a prostate segmentation method, which combines shear transformation and deformable convolution. The method in this paper solved the problem of inaccurate segmentation results caused by asymmetric and irregular shapes of the prostate in TRUS images and fuzzy edges. By replacing the original convolution with improved convolution-assisted network training in U-Net segmentation network, more continuous boundary information was extracted, so as to be more sensitive to image boundary information. The effectiveness of the proposed model was verified by conducting comparison experiments with other competitive methods. Furthermore, the results showed that it was closer to the ground truth and the segmentation effect of the boundary region was more accurate. However, some possibilities still remain to be explored. In the near future, more effort will be invested into exploring the causes of unclear boundary regions in image segmentation.

## Figures and Tables

**Figure 1 biomedicines-11-00646-f001:**
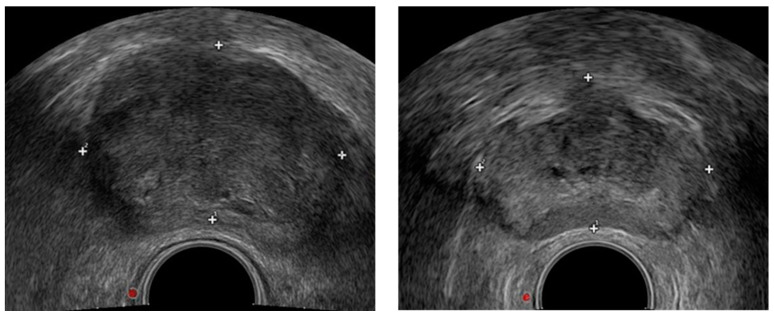
TRUS imaging of prostate.

**Figure 2 biomedicines-11-00646-f002:**
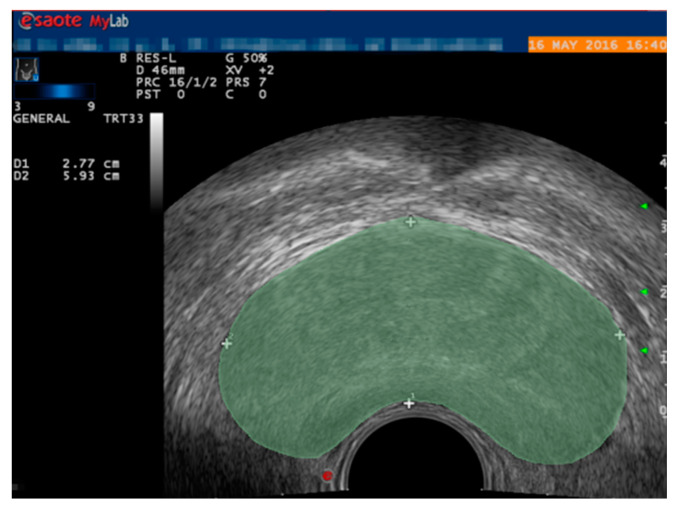
TRUS image of the prostate.

**Figure 3 biomedicines-11-00646-f003:**
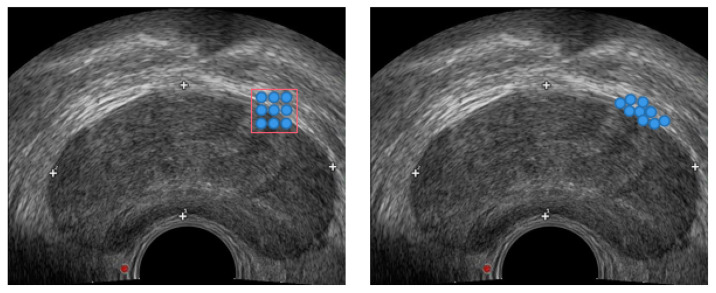
Basic convolution and deformable convolution sampling graph with convolution kernel size 3 × 3.

**Figure 4 biomedicines-11-00646-f004:**
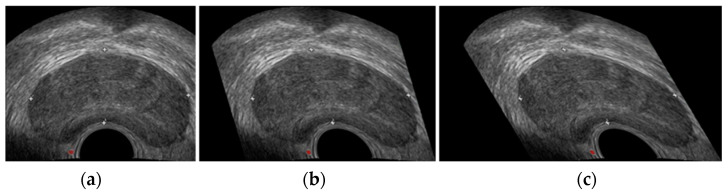
Effect diagram of shear transformation. (**a**) is the original image, (**b**) shows the shear transformation with a transverse parameter of 0.3, (**c**) shows the shear transformation with a transverse parameter of 0.6).

**Figure 5 biomedicines-11-00646-f005:**
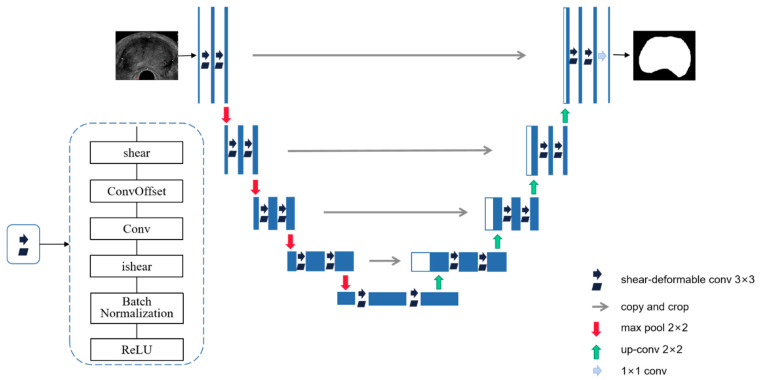
Structure diagram of prostate segmentation network model based on DSU-Net.

**Figure 6 biomedicines-11-00646-f006:**
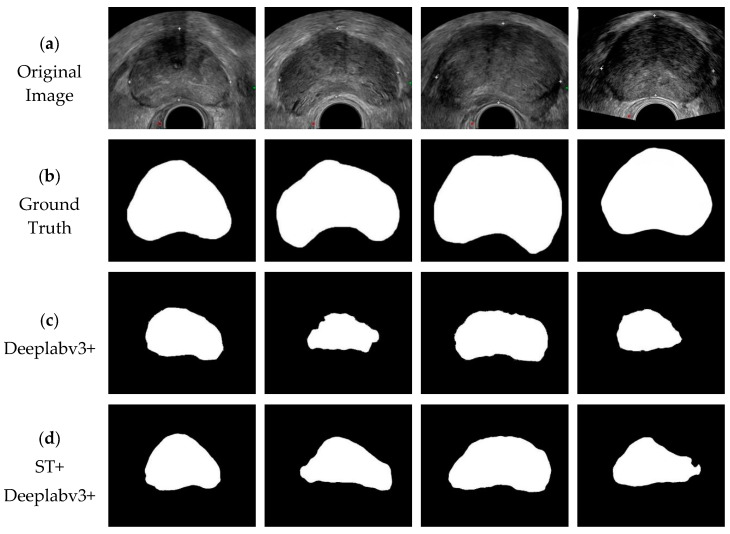
Visualized results of the segmentation experiment comparison method.

**Table 1 biomedicines-11-00646-t001:** Experimental parameter table.

Learning Rate	Training Time	Batch Size	Optimizer
0.001	80 epochs	8	Adam

**Table 2 biomedicines-11-00646-t002:** Experimental results of prostate segmentation in TRUS images.

Method	Dice	Jaccard
DeepLabv3+	0.750	0.605
ST+DeepLabv3+	0.784	0.648
PSPNet	0.879	0.787
ST+PSPNet	0.895	0.834
U-Net	0.923	0.886
ST+U-Net	0.935	0.890
DU-Net	0.941	0.893
DSU-Net	**0.957**	**0.925**

## Data Availability

The dataset is privately owned cannot be released.

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
