# Peer review of "Prostate Ultrasound Image Segmentation Based on DSU-Net"

_biomedicines, 2023, doi:10.3390/biomedicines11030646_

Round 1
Reviewer 1 Report
This paper proposed a deep learning based method for prostate segmentation in TRUS images. The proposed method applied a deformable convolution kernel into the U-Net architecture, and achieved a better segmentation accuracy when using 1057 2D TRUS images in terms of Dice. The methodology looks interesting on this particular application although both technical and clinical values are limited. My major concerns exist due to:
1) More thorough literature review on this topic is needed. What’s surprising me is many latest papers from the groups, such as Dr. Aaron Fenster, Dr. Anant Madabhushi, Dr. Baowei Fei, etc, are missing
2) The latest prostate TRUS segmentation accuracy in literature should be addressed. It may help to better evaluate the scientific and clinical values of the paper.
3) Given the deformable convolution has been proposed and applied in many other fields, it would be suggested to extensively evaluated the performance, such as using different metrics, publicly available datasets, e.g., PROMISE 2012. Since the proposed network is a general segmentation framework, and not restricted in this application, more evaluations on other application would make the paper much stronger.
4) Both technical and Clinical values of the paper should be discussed, given many papers on this application have been published.
5) It is not clear what the Dice values in Table 2 mean? Mean or Median? Segmentation variance is also important in addition to this. In addition, the Dice metric may be biased by prostate volume. Distance errors should be also reported.
6) Extensive editing of the language is required.
Author Response
1) There are no relevant papers about Dr. Aaron Fenster, Dr. Anant Madabhushi, Dr. Baowei Fei in my thesis literature. I wonder if there is some misunderstanding
2) Thank you very much for your suggestion. The method I proposed in this article can also perform segmentation work very well.
3) In this paper, I used the prostate ultrasound imaging data provided by Shanghai Ruijin Hospital, which belongs to the privacy of patients. Through verification, it is found that the method in this paper has achieved good results on these data sets
4) Comparing the segmentation model and basic model in this paper with the basic model with shear transformation and the model based on deformable convolution, it can be seen from the quantitative indicators and visualization results that the segmentation results of this paper are more suitable than the comparison model Real labels, the segmentation effect of the boundary area is also more accurate.
5) The value of Dice has been explained in Section 3.2
6) I am working on the point you mentioned.
If there is any problem, I would appreciate your correction.
Hope you have a nice day.
Reviewer 2 Report
Due to the asymmetric shape and blurred boundary line of prostate in TRUS images, it is hard to segment the accurate segmentation results by existing segmentation methods. Therefore, this paper proposes a prostate segmentatrion method, which combines shear transformation and deformable convolution.
The review comments are as follows:
- Since this is a human subject study, IRB review is required. If you get that, you should fill in Institutional Review Board Statement, Informed Consent Statement, and Data Availability Statement.
- Explain in detail how you obtained the test images. Additionally how did you get the ground truth?
- There is few explanation in data pre-processing, usage of Shear Transformation, and so on.
Author Response
response 1:The meaning of dice is explained in section 3.2
The remaining content to be added has been completed, please see the attachment for details (updated manuscript)
Round 2
Reviewer 1 Report
1) There are no relevant papers about Dr. Aaron Fenster, Dr. Anant Madabhushi, Dr. Baowei Fei in my thesis literature. I wonder if there is some misunderstanding.
Please carefully check the papers about the TRUS segmentation from those groups.
https://scholar.google.com/citations?hl=en&user=9VIOS1wAAAAJ&view_op=list_works&sortby=pubdate
2) Thank you very much for your suggestion. The method I proposed in this article can also perform segmentation work very well.
Please qualitiatively review the accuracy of TRUS prostate segmentation
3) In this paper, I used the prostate ultrasound imaging data provided by Shanghai Ruijin Hospital, which belongs to the privacy of patients. Through verification, it is found that the method in this paper has achieved good results on these data sets
In addition to the evaluation on your internal dataset, evaluations on public dataset should make your results more convicing.
4) Comparing the segmentation model and basic model in this paper with the basic model with shear transformation and the model based on deformable convolution, it can be seen from the quantitative indicators and visualization results that the segmentation results of this paper are more suitable than the comparison model Real labels, the segmentation effect of the boundary area is also more accurate.
What about your clinical values? there is any added clinical values on the exsiting prostate segmentation methods? In another words, how your marginal improvement changes the prostate intervention clinically?
Author Response
Response 1: Please provide your response for Point 1. (in red)
Thanks for your advice, I have shortened the content of Part 2&3, and all my formulas have been marked
Response 2: Please provide your response for Point 2. (in red)
Thank you for your advice
Regarding the formulas for the two metrics defined in my paper, I have explained how they are calculated and what each letter in each formula means to help readers understand
Round 3
Reviewer 1 Report
I do not have any comments
Author Response
-